Published in FAST Workshop on Smalltalk Related Technologies (11/2022)

# PowerlangJS: A Quick Way to Get Your Smalltalk to the Web?

**Javier Pimás**                                                                                               *javierpimas@gmail.com*
*Universidad de Buenos Aires*

**Reviewed on OpenReview:** *https://openreview.net/forum?id=DRUVWUDX_z*

## Abstract

Powerlang bootstrapper was designed to bring alive Smalltalk systems from specifications stored in source code files. Powerlang includes an evaluator that is able to execute initialization code of the Smalltalk being bootstrapped. Could we use that same evaluator for running that same Smalltalk on the Web? We present our work in progress towards that goal, which involves transpiling the Smalltalk evaluator, written in Smalltalk, to JavaScript, generating a Smalltalk image compatible with that evaluator, and optimizing the evaluator and the Smalltalk code of the image.

## 1 Introduction

Smalltalk was created decades before the Web. When the Web became popular, Smalltalk implementations added features that let them interact with it, by issuing or serving requests in some form of client/server communication. Later on, after JavaScript was created and evolved, it became possible to implement Smalltalk systems on top of it.

Amber (Mason, 2015) is a Smalltalk-alike programming environment that runs on top of JavaScript. Amber aligns Smalltalk objects with their JavaScript counterparts as much as possible, to allow a smooth interaction between JS and ST worlds. For example, Smalltalk closure objects get converted to JavaScript functions and can be invoked from JavaScript code. Smalltalk code is *transpiled* to JavaScript by Amber's compiler. Its performance is at least two orders of magnitude slower than native JavaScript code.

SqueakJS (Freudenberg et al., 2014) is a JavaScript implementation of the Squeak object and execution models (Ingalls et al., 1997). In SqueakJS it is basically possible to take a binary Squeak image as-is and to load it into a web page, as it provides both an image reader that loads the objects of the image, converting them in JavaScript objects and also an interpreter for Squeak's bytecodes written in JS. SqueakJS is not using the Slang-based Squeak Virtual Machine (VM) because that VM operates directly with words on memory, which is too low-level and inappropriate to transpile to JavaScript.

PharoJS (Bouraqadi & Mason, 2018) is a framework that supports developing applications in Pharo while transpiling parts of them to JavaScript in order to let them run on top of a JS engine.

Newspeak (Bracha et al., 2008) is an object-capability programming platform that lets developers program in web browsers. Its VM is written in C++, and transpiled to WebAssembly trough Emscripten (Zakai, 2011).

While we can find many attempts to bring Smalltalk closer to the Web, they are all ad-hoc implementations, targeted for different scenarios. Currently, there is not a general technique to create a JavaScript-based VM for any Smalltalk dialect. The contribution of this work is to present such a mechanism. We propose generating Smalltalk VMs that run on top of JavaScript by transpiling Smalltalk evaluators written in

Smalltalk into JavaScript. We show PowerlangJS, a research experiment that takes this approach to let Bee Smalltalk run on the Web.

## 2 Approach

Our proposed approach is based in two initial insights:

**Low-level nature of Smalltalk Interpreters.** Smalltalk interpreters written in Smalltalk are pieces of code that can be written without using the advanced features that Smalltalk provides. For example, the Squeak VM is written in Slang, which can be translated to C thanks to that property. It does not make use of #doesNotUnderstand, exceptions or reflection during execution of Smalltalk code. This makes it easier to transpile the code of an interpreter to other languages than generic Smalltalk code that may make use of Smalltalk high-level dynamic features.

**Smalltalk message dispatch can be optimized by interpreters.** Inline caching (Deutsch & Schiffman, 1984; Hölzle et al., 1991) is the most important optimization in dynamically typed languages. While it is usually implemented at JIT level, it does not require to be so, as shown in Cola (Piumarta & Warth, 2008), Quickening (Brunthaler, 2010), and Truffle (Würthinger et al., 2012). An interpreter can implement the full Smalltalk message-send semantics (with #doesNotUnderstand) and use inline caching to optimize the common cases.

These two insights lead to a generalizable approach to implement Smalltalk VMs that can run in JavaScript. Instead of writing a Smalltalk VM in JavaScript, we propose writing an evaluator in Smalltalk, and then transpiling it to JavaScript.

Our approach takes a Smalltalk dialect to be run, and then the following three main elements have to be implemented:

**Smalltalk AST Evaluator.** An AST visitor that is capable of evaluating the code of the Smalltalk dialect, with a message dispatch mechanism that uses inline caching.

**Glue Code.** A series of handwritten JS functions and classes that mimic the behavior of Smalltalk core libraries used by the evaluator, like Booleans, Arrays, Strings and Dictionaries. The glue-code only needs to provide support for the functionality used by the Smalltalk Evaluator.

**Smalltalk-to-JavaScript Transpiler.** A transpilation tool that is able to generate JS code from the classes that represent the evaluator.

The approach consists of transpiling the Smalltalk Evaluator with the Smalltalk-to-JavaScript transpiler. Each class of the Evaluator is mapped to an equivalent JS class. Core Smalltalk classes are not transpiled, because using transpiled versions of Booleans, Arrays, Strings or Dictionaries would be inefficient compared to native JS versions of the same concepts. Instead, the glue code augments the JS classes with the required functionality by adding methods to their prototypes. The transpiler is in charge of converting literals and to their corresponding JS versions. At initialization time, the glue code loads an image file and for each Smalltalk object it creates a JS object, to then just startup the evaluator.

### 2.1 PowerlangJS

To validate our approach, we built PowerlangJS, which is a direct derivation of Powerlang. Powerlang is a bootstrapping framework that includes an interpreter to allow generating Smalltalk images from Smalltalk source code stored in files. We now describe Powerlang to provide a better context for the reader.

Powerlang contains two main components: an image bootstrapper and an evaluator. The image bootstrapper reads the source code files and creates a virtual image with the objects that are the bare bones required for bootstrapping: instances of classes, metaclasses, nil, true and false are created and linked together according

to the typical metacircular design of any Smalltalk. After this initial linking, many parts of the system have to be initialized: constants, pool and class variables, globals, etc. While this initialization could be done by the bootstrapper, we chose instead to leave the bootstrapped system to initialize itself instead. This allows to place more of the code related to the bootstrapped system within the boundaries of that same system.

In order to allow the system initializing itself, the evaluator is used to send #initialize to a root object such as Smalltalk, which can then perform all the initialization process. To save the image, the evaluator can be used again: we just need to send #saveImage to the Smalltalk object.

## 2.2 Generating a JavaScript Platform

The bootstrapped system can save the image in the format that it needs for the VM it expects to run on. Unlike in traditional Powerlang, or in SqueakJS, PowerlangJS generates a different image format crafted for the Web: instead of serializing objects of the image in a binary format, we generate a JSON file with them that can be directly parsed by the JS engine[1].

This image contains classes, methods and objects of a Smalltalk kernel. It is made of 10,000 objects and occupies around 900kb.

### 2.2.1 Transpiling a Smalltalk Subset to JavaScript

To be able to execute Smalltalk code on top of JS, a Smalltalk evaluator made of JS code is needed. Our transpiler takes the classes and methods of the evaluator, implemented in Smalltalk, and converts them to JavaScript. For each class in Smalltalk, it generates a JS class. Since it only needs to transpile a specific corpus of code -the Smalltalk evaluator- it does not need to care about support features like exceptions, reflection or doesNotUnderstand, and we can rename selectors and instance variables. This simplifies the transpiler design, compared to approaches that aim for maximum compatibility of general Smalltalk code. It also allows for more straightforward code, which can help JavaScript JIT compilers to optimize it. For example, when a selector is modified during transpilation, no care needs to be taken that #perform keeps working, because the evaluator does not send it.

Instance variables, temporaries, method arguments and selectors are renamed to avoid clashes in JavaScript world. Since in JavaScript it is not possible to have instance variables with the same names as methods, all instance variable names are added the prefix _. Also, it is not possible to use : in method names, so we rename selectors replacing each : with _. These naming conventions avoid any ambiguity when transforming code. For example, the method

```
Point>>#x
    ^x
```

would be translated to `function x() { return _x; }`, while the method

```
Point>>#x: anInteger
    x := anInteger
```

becomes `function x_(anInteger) { _x = anInteger; return this; }`.

Block closures are translated to JavaScript arrow functions. As JS does not support non-local returns, we simulate them with exceptions: methods that contain non-local returns have their code surrounded in a try-catch block. Before the try statement starts, a *home* object is created. On each non-local return the home object is thrown, paired with a result. The catch block looks for a match with the home object and if identical, it returns the paired result, re-throwing the exception in any other case. As an example, we show the translation of a method that includes a non-local return:

---

[1]As JSON only allows to write data trees in contrast to data graphs, we perform a serialization step to write references as indexes in a table, breaking circular references, and we replace indexes back with direct references when deserializing.

```
nonLocal
    value := dictionary at: key ifAbsent: [^1].
    ^value
```

which gets converted to:

```
function nonLocal()
{
    let _home = [];
    try {
        let value = nil;
        value = dictionary.at_ifAbsent_(key, () => { home[0] = 1; throw home; });
        return value;
    }
    catch (e)
    {
        if (e === home) return e[0] else throw e;
    }
}
```

### 2.2.2 Glue-code library

Besides objects implemented *in* the evaluator, the evaluator itself uses objects that belong to the core of Smalltalk: nil, true, false, numbers, strings, closures, arrays, dictionaries, intervals and streams. Instead of also transpiling the methods of those classes, we just took what JavaScript base library provides and augmented it with methods that mimic the API expected by Smalltalk code. For example, in Boolean's JavaScript prototype an equivalent to ifTrue:ifFalse: is added like this:

```
Boolean.prototype.ifTrue_ifFalse_ = function(closureTrue, closureFalse) {
    if (this) { return closureTrue() } else { return closureFalse() }
}
```

The only object that is not possible to extend like this was nil, because JavaScript's equivalent null does not support adding any extension method. For that reason, Smalltalk's nil was implemented in JavaScript as a global object named nil.

With this kind of glue code we were able to generate a compatibility library for the transpiled evaluator in less than 150 methods. This includes:

**Extensions to JS classes.** Boolean, Array, Map, String and Function.

**Compatibility classes.** ReadStream, WriteStream, Interval and Stretch.

**Debugging code.** Mainly toString methods, which are JavaScript's equivalents to Smalltalk's printString

The most important parts this code can be found in Appendix A.

## 3 Evaluation

The approach has the benefit that the evaluator can be completely developed in Smalltalk.

Unlike Amber, the resulting Smalltalk is not tied to the JavaScript semantics. While aligning ST objects to "mirror" JS objects can be useful to reduce interoperability barriers, it can also be problematic as it poses

restrictions to the way Smalltalk objects are designed. For example in Amber, as in JavaScript, there are no True and False classes, just Boolean.

Compared to PharoJS, the transpiling process of PowerlangJS is simpler, because the only code that needs to be transpiled is the one of the evaluator, which does not use any advanced Smalltalk feature, like reflection, metaprogramming or exceptions. PharoJS can deal with more complex Smalltalk code, but at the cost of a more complex transpilation. To give an example, to be able to support #doesNotUnderstand in code transpiled to JavaScript, while at the same time using JS method dispatch, it has to add a method in Object prototype for existing symbol in the system.

WebAssembly/EmScripten based approaches allow obtaining maximum compatibility with the original VM. While Wasm code is executed efficiently, the whole system is still limited to the efficiency of the original VM. A slow VM running on top of WebAssembly will still run slow, while writing a fast VM takes great efforts. Our approach, on the other hand, takes advantage of the optimization capabilities of the JS engine. Instead of requiring to implement low-level optimizations, we optimize high-level operations and leave the rest to the JS JIT.

The application programmer working on the system does not see JavaScript, because there is no transpilation of application code to JS. The only JavaScript required is the VM and the glue code, which the application just needs to include, and the evaluator is fully developed in Smalltalk.

There is no need to write a Garbage Collector, as Smalltalk objects are instantiated through JavaScript objects and get collected automatically. The FFI interface is a Smalltalk to JavaScript one.

## 3.1 Debugging

The system has different debugging tools depending on which component is being debugged. The Smalltalk evaluator and the transpiler are written in Smalltalk and can be developed with traditional Smalltalk tools, we currently use Pharo.

After transpilation of the evaluator, the resulting code is a set of classes that look pretty similar to their Smalltalk equivalents. These classes correspond to the evaluator, not to the target Smalltalk image. The code of the VM is run on top of a JS engine, like V8, and loads a Smalltalk image. To debug both the transpiled VM and the Smalltalk code inside it, we are just using the JavaScript debugger for now. The glue code includes 100 lines of code to allow showing the Smalltalk objects in JavaScript in a similar way than when using a Smalltalk inspector[2]. In the future, we plan to implement a Webside (Amaral, 2022) backend to be able to work in a more traditional way with the Smalltalk image.

## 3.2 Performance

In Powerlang the evaluator is implemented as a visitor on a low-level version of compiled-method ASTs called *treecodes*. Treecodes are encoded as byte arrays (like conventional bytecodes) and stored on compiled methods. They get decoded on-the-fly by the evaluator. AST visiting evaluators have the benefit that they can be written in very few lines of code. Currently, Powerlang evaluation framework contains less than 2500 lines of code.

Because the evaluator itself is a visitor, the execution of message sends is done recursively, unlike bytecode evaluators where execution follows a fetch-decode-execute non-recursive loop. Truffle shows that recursive execution is well suited for optimization in method-JIT optimizing compilers found in Java VMs, because these VMs are good at dynamically inlining the recursive calls, and then optimizing away many of the abstractions used by the evaluator. These are the same kinds of optimizing compilers present in most JS engines, so we expect to obtain similar benefits when running on JS instead of Java.

To take more advantage of recursive execution, our approach uses a high-level implementation of inline caching. The evaluator keeps track of the results of message lookup, and implements a global cache and a polymorphic inline cache. Overall, the approach balances high-level Smalltalk optimizations with the

---

[2]JavaScript provides toString as an analogous to Smalltalk's #printString.

optimization stages of JS engines: the ST message lookup, which is hard to optimize by a JS engine, is optimized by the Smalltalk evaluator, while method inlining, adaptive recompilation and other low-level algorithms are left to the JS engine.

While our approach is similar to Truffle, our evaluator is written in the *guest* language, Smalltalk, while in Truffle the evaluator is written in the *host* language, which in Truffle's case is Java. TruffleSqueak (Niephaus et al., 2018) is an implementation of Squeak that uses Truffle. Another difference with Truffle is that our approach does not inline the AST of the message sends into the senders, as Truffle does, although that can be implemented in the future, and can also be done in Smalltalk.

When executing PowerlangJS on top of V8 JS engine we initially noticed that the VM did not perform as fast as expected. Transpiling methods written in a high level fashion, such as the way explained for ifTrue:ifFalse: which generates closures, or using exceptions for implementing non-local returns, seems to be too much for V8, which is not able to optimize the code accordingly. However, our approach has the benefit that code of the Smalltalk VM can be pre-processed before generating the JS version, in order to optimize the generated JS beforehand. Thanks to this, we were able to generate more efficient transpiled code, letting V8 execute PowerlangJS methods faster, and the whole Smalltalk environment in general.

While obtaining a full assessment of performance was not in our goals, we did run some Fibonacci benchmarks. The first runs showed our approach being more than 25 times slower than Amber. After some very simple optimizations (basically changing the transpiled code to avoid using arrow functions on branches like ifTrue:, ifFalse:, ifNil:, whileTrue:, etc, we could get performance parity with Amber. Later experiments involving more complex transpilation approaches showed evidence that there is still room to get at least another 10x improvement in speed.

### 3.3 Current status

PowerlangJS is currently able to execute most Smalltalk code, except for operations on floating point numbers and large integers. Our experiments were conducted on top of a modified, open-source version of Bee Smalltalk, which includes only a Smalltalk-80 alike kernel. We are working on improving the transpiler to allow for automatic inlining of JS functions (specially the glue code), to make the final transpiled JS code look like ideomatic JS and does not use JS lamdas for branching and cycling, because it hinders performance.

## 4 Conclusions

We have shown a technique to implement a Smalltalk VM that can run on top of a JavaScript engine, and evaluate the very same Smalltalk code developed for a traditional VM, without any modification. This technique allows to maintain the full semantics of the Smalltalk code: classes, metaclasses, closures, nil, true, false... all ST objects maintain their semantics. Instead of transpiling the ST code of the application, which is hard, we implement a metacircular Smalltalk evaluator and transpile its code to JavaScript, which is simpler. Our initial benchmarks suggest that the approach runs as fast as other available Smalltalk-in-JavaScript implementations and promises to run faster after some more optimization. While targeted at Smalltalk, we believe that the same technique could also be applied to implement other programming languages on top of JavaScript.

## 5 Future Work

The current PowerlangJS transpiler is a Smalltalk AST visitor translating directly to JavaScript code. While simple, it makes hard to perform optimizations on the code. We started working on a transpiler that goes from Smalltalk AST to a simplified JavaScript AST, but noticed some drawbacks. It is harder to perform some optimizations (such as inlining) in an intermediate representation that lacks the SSA property (Alpern et al., 1988). It might be easier to take advantage instead of the optimizing compiler we already have in Bee. We could then transform Smalltalk AST to an optimizing SSA-based IR, then optimize and, finally, to transform to JavaScript code.

**Acknowledgments**

The author wants to thank Carlos Ferro and Guillermo Amaral of Quorum Software for providing valuable feedback on this work.

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

## A   Appendix

```javascript
globalThis.nil = {}

globalThis._cascade = function(receiver, cascadeStatements) {
    return cascadeStatements(receiver);
}

Object.prototype.ifNil_ = function(closure) {
    if (this == nil) { return closure() } else { return this }
}

Object.prototype.ifNotNil_ = function(closure) {
    if (this == nil) { return nil } else {return closure(this)}
}
Object.prototype.ifNil_ifNotNil_ = function(closureNil, closureNotNil) {
    if (this == nil) { return closureNil() } else {return closureNotNil(this) } }

Object.prototype.isNil = function() { return (this === nil) }
Object.prototype.notNil = function() { return (!this.isNil()) }

Object.prototype.value = function () { return this; }

Object.prototype.eval_ = function(string) {
     return eval(string) }

Object.prototype.isSmallInteger = function(value) { return Number.isInteger(value); }

Object.prototype.isCollection = function() { return false; }
Array.prototype.isCollection  = function() { return true; }
Map.prototype.isCollection    = function() { return true; }

Object.prototype.initialize = function() { return this;}
Object.prototype.basicNew = function() { return new this; }
Object.prototype.new = function() { const obj = new this; obj.initialize(); return obj; }
Object.prototype.class = function() { return this.constructor; }
```

```javascript
// add some Smalltalk-ish methods to JS booleans

Boolean.prototype.ifTrue_  = function(closure) {
    if (this) { return closure() } else {return nil }
}
Boolean.prototype.ifFalse_ = function(closure) {
    if (this) { return nil } else { return closure() }
}
Boolean.prototype.ifTrue_ifFalse_ = function(closureTrue, closureFalse) {
    if (this) { return closureTrue() } else {return closureFalse() }
}

Boolean.prototype.or_  = function (closure) { return this || closure() }
Boolean.prototype.orNot_  = function (closure) { return this || !closure() }
Boolean.prototype.and_ = function (closure) { return this && closure() }
Boolean.prototype.andNot_ = function (closure) { return this && !closure() }
Boolean.prototype.not = function () { return !this }

Array.new_ = function(size) { return new Array(size); }
Array.new_withAll_ = function(size, value) { return Array(size).fill(value); }
Array.with_with_ = function(first, second) { return [first, second]; }
Array.prototype.size  = function()        { return this.length; }
Array.prototype.isEmpty  = function()         { return this.length == 0; }
Array.prototype.asArray  = function()         { return this; }
Array.prototype.at_     = function(index)        { return this[index-1]; }
Array.prototype.at_put_ = function(index, object) { return this[index-1] = object; }
Array.prototype.atAllPut_ = function(value) { return this.fill(value); }
Array.prototype.allButLast = function() { return this.slice(0,-1); }
Array.prototype.first    = function()        { return this[0]; }
Array.prototype.second   = function()        { return this[1]; }
Array.prototype.last     = function()        { return this[this.length-1]; }
Array.prototype.asString = function() { return String.fromCharCode.apply(null, this); }
Array.prototype.do_ = function(closure) {
    this.forEach(closure);
}
Array.prototype.withIndexDo_ = function(closure) {
    this.forEach((element, index) => {closure(element, index + 1)} );
}
Array.prototype.do_separatedBy_ = function(closure, separated) {
    this.forEach((value, index) => {
        closure(value);
        if (!Object.is(this.length - 1, index)) {separated()}
    });
}

Array.prototype.add_  = function(object) { return this.push(object); }
Array.prototype._comma = function(array) { return this.concat(array); }
Array.prototype._equal = function(value) {
    return Array.isArray(value) &&
        this.length === value.length &&
        this.every((val, index) => val === value[index]);
}
```

```javascript
Map.withAll_ = function(associations) {
    const result = new this;
    associations.forEach((assoc) => result.set(assoc.key(), assoc.value()));
    return result;
}

Map.prototype.at_           = function(key)         { return this.get(key); }
Map.prototype.at_put_       = function(key, value) { return this.set(key, value); }
Map.prototype.at_ifPresent_ = function(key, closure) {
    if (this.has(key))
        return closure(this.get(key));
    else
        return nil;
}

Map.prototype.at_ifAbsentPut_ = function(key, closure) {
    let v = this.get(key);
    if (v)
        return v;
    else
        v = closure();
        this.set(key, v);
    return v;
}

Map.prototype.removeKey_ifAbsent_ = function(key, closure) {
    if (!this.delete(key)) { closure();}
}

Map.prototype.keysAndValuesDo_ = function(closure) {
    this.forEach((value, key) => closure(key, value));
}

String.prototype.asSymbol = function() { return this; }
String.prototype.asByteArray = function() {
    return Array.from(this).map((char) => char.charCodeAt(0));
}
```

```javascript
// ~~~~~~~~~~~~~~~~~~~ Block Closures ~~~~~~~~~~~~~~~~~~~~~~~~

Function.prototype.value = function () { return this(); }
Function.prototype.value_ = function (a) { return this(a); }
Function.prototype.value_value_ = function (a, b) { return this(a, b); }
Function.prototype.value_value_value_ = function (a, b, c) { return this(a, b, c); }

// loop helpers
Function.prototype.whileTrue = function () {
    while(this()) { }
    return nil;
}

Function.prototype.whileFalse = function () {
    while(!this()) { }
    return nil;
}

Function.prototype.whileTrue_ = function (block) {
    while(this()) { block() }
    return nil;
}

Function.prototype.whileFalse_ = function (block) {
    while(!this()) { block() }
    return nil;
}

Object.prototype._arrow = function(value) {
    const t = this;
    return {
        key: function() {return t; },
        value: function() {return value; },
        key_: function(k) {this.key = function() { return k; }; return this;},
        value_: function(v) {this.value = function() { return v; }; return this;}
    };
}
```

```javascript
Object.prototype._equal = function(value) { return this ==  value; }
Object.prototype._notEqual = function(value) { return this !=  value; }
Object.prototype._equalEqual = function(value) { return this ===  value; }

// add some Smalltalk-ish methods to JS numbers
Number.prototype._lessEqualThan = function(value) { return this <=  value; }
Number.prototype._lessThan = function(value) { return this <  value; }
Number.prototype._greaterEqualThan = function(value) { return this >=  value; }
Number.prototype._greaterThan = function(value) { return this >  value; }

Number.prototype._plus = function(value) { return this + value; }
Number.prototype._minus = function(value) { return this - value; }
Number.prototype._times = function(value) { return this * value; }
Number.prototype._slash = function(value) { return this / value; }
Number.prototype._modulo = function(value) { return this % value; }
Number.prototype._integerQuotient = function(value) { return Math.floor(this / value); }
Number.prototype._and = function(value) { return this & value; }
Number.prototype._or = function(value) { return this | value; }
Number.prototype.bitAnd_ = function(value) { return this & value; }
Number.prototype.bitOr_ = function(value) { return this | value; }
Number.prototype._shiftLeft = function(value) { return this << value; }
Number.prototype._shiftRight = function(value) { return this >> value; }
Number.prototype.anyMask_ = function(value) { return (this & value) != 0; }
Number.prototype.noMask_ = function(value) { return (this & value) == 0; }

Number.prototype.timesRepeat_ = function(closure) {
    for (let i = 0; i < this; i++) { closure(); }
}
Number.prototype.to_do_ = function(limit, closure) {
    for (let i = this; i <= limit; i++) { closure(i); }
}
Number.prototype.to_by_do_ = function(limit, increment, closure) {
    if (increment > 0)
        for (let i = this; i <= limit; i=i+increment) { closure(i); }
    else
        for (let i = this; i >= limit; i=i+increment) { closure(i); }
}
```

```javascript
// ~~~~~~~~~~~~~~~~~~~ Interval ~~~~~~~~~~~~~~~~~~~~~~~~

let Interval = class {
    constructor(start, end, step = 1) {
        this.start = start; this.end = end; this.step = 1
    };

    at_(anInteger) {
        if (anInteger > 0) {
            const result = this.start + (anInteger - 1 * this.step);
            const min = Math.min(this.start, this.end);
            const max = Math.max(this.start, this.end);
            if (min <= result && result <= max) return result;
            if (anInteger == this.size() ) return this.end;
        }

        throw "outOfBoundsIndex";
    }

    do_(closure) {
        if (this.step > 0) {
            for (let i = this.start; i <= this.end; i=i+this.step)
                closure(i)
        } else {
            for (let i = this.start; i >= this.end; i=i+this.step)
                closure(i)
        }
    }

    collect_(closure) {
        const result = [];
        const s = this.size();
        for (let i = 1; i <= s; i++)
            result.push(closure(this.at_(i)));
        return result;
    }

    size() {
        let _size = Math.max(0, Math.floor((this.end - this.start) / this.step) + 1);
        const x = this.step * _size + this.start;
        if ((this.step < 0 && this.end <= x) || (this.step > 0 && x <= this.end))
            _size = _size + 1;
        return _size;
    }
}

Number.prototype.to_ = function(value) {
    return new Interval(this, value);
}
Number.prototype.to_by_ = function(limit, increment) {
    return new Interval(this, limit, increment);
}
```

