# OpenReview forum: "PowerlangJS: A Quick Way to Get Your Smalltalk to the Web?"
_FAST.org.ar/2022/Workshop — FAST Smalltalk 2022_

### Official Review · Reviewer_SZdy · 2022-10-20
**Interesting and good discussion launcher**

**Rating:** 7
**Confidence:** 5

**Review:**

Thanks, I liked reading the paper, having different implementation strategies is a good way to learn about different trade-offs.

# Paper Summary

The paper discussed PowerLangJS, a Smalltalk implementation for the web, based on PowerLang.
The key idea is to make Smalltalk programs execute on the web by producing a Smalltalk evaluator (an interpreter) running on Javascript.
The evaluator is obtained by transpiling a Smalltalk written evaluator to Javascript.
The paper explains some of the core design decisions of their transpiler.
Finally, it compares with related work and informally evaluates the debuggability and performance of the approach.

# Comments

The paper is interesting to read, short, and right to the point.
I like the idea of translating the evaluator instead of translating a program.

I wonder if you tried to translate your evaluator using PharoJS and you failed.
The "having a more complex transpilation process" is a strange reason for not using it.
We do not all write our own C compilers every day, and that's fine :).
It would be nice in the comparison to see how PharoJS translates non-local returns and deal with name collisions.

I found the discussion on performance a bit too informal for my taste.
- I was very (and very positively) intrigued by the treecodes.
  Is there a ref to add to the paper? a link? Is it related somehow to the idea of slim binaries?
- The claim that VMs are better at inlining recursive calls, implies somehow that they are better at that than... ? I assume than optimizing loops? But I would rather believe the opposite. I would be interested to see a ref for such a claim. Otherwise, my understanding is that an interpreter will be optimized poorly (or at least not take advantage of the full JS optimizer) unless doing some more complex stuff such as partially evaluating methods to create specialized interpreters for particular methods.
- Current work on truffle AST interpreters show that they are slow without partial evaluation, and optimizing them is a hard task nowadays even on top of mature JVMs. Check for example the work on super nodes from S. Marr et al.
- I found it a bit disappointing that the fourth paragraph of the performance section says that to achieve the reasonable performance you had to "remove" the extensions you explained in section 2.1.2, it's like you're invalidating your own paper! Probably you want to rewrite it differently ^^
- Finally, I remain very curious of what are those "more complex transpilation approaches" to each that 10x!!

---

### Official Review · Reviewer_aW6M · 2022-11-01

**Rating:** 6
**Confidence:** 3

**Review:**

This short paper is about an approach to bootstrapping a Smalltalk image, from source, as a JavaScript payload that can run in a web browser. Unlike the SqueakJS approach, it avoids supporting an existing image format, but appears to gain in minimality and simplicity by translating only a minimal interpreter and bootstrap objects, and allowing the image to bootstrap itself from there as part of its initialization.

I believe this is interesting work, but the paper is rather preliminary and tends to over-claim a little. I find it difficult to wholeheartedly recommend, but it might still be a fair addition to the program.

The paper claims to be general in ways that other techniques, such as SqueakJS, are not. However, I could not convince myself that this is true. Can the claim (on p1) that it works for "any Smalltalk implementation" really be true?

Much of the early part of section 3 (Evaluation) is really a design discussion, and belongs in the Introduction. It is generally clearer than the Introduction about explaining the design points.

There seems to be a strong parallel with GraalSqueak (of Fabio Niephaus and others), which uses the Graal/Truffle infrastructure analogously to how V8 or other JavaScript engines are used here. Is that a fair comparison? Some discussion would help.

I was curious about how much bigger the JSON images were, in bytes, compared to a Squeak-format image or other binary format.

I was also interested to learn how compatible the glue code library (section 2.1.2) had so far been made. What are the largest Smalltalk programs/codebases that can be run? What supports the Conclusion's claim that "all ST objects maintain their semantics"?

Is there a reason why the paper says 'evaluator' and not 'interpreter'? The latter would be more specific and therefore clearer.